# Weakly-Supervised Learning of Visual Relations in Multimodal Pretraining

**Emanuele Bugliarello**[*,ð,c]   **Aida Nematzadeh**[‡,ð]   **Lisa Anne Hendricks**[‡,ð]

[ð]Google DeepMind   [c]University of Copenhagen

## Abstract

Recent work in vision-and-language pretraining has investigated supervised signals from object detection data to learn better, fine-grained multimodal representations. In this work, we take a step further and explore how we can tap into supervision from small-scale visual relation data. In particular, we propose two pretraining approaches to contextualise visual entities in a multimodal setup. With *verbalised scene graphs*, we transform visual relation triplets into structured captions, and treat them as additional image descriptions. With *masked relation prediction*, we further encourage relating entities from image regions with visually masked contexts. When applied to strong baselines pretrained on large amounts of Web data, zero-shot evaluations on both coarse-grained and fine-grained tasks show the efficacy of our methods in learning multimodal representations from weakly-supervised relations data.

## 1 Introduction

Current vision-and-language models (VLMs) are pretrained on large amounts of image–text pairs collected from the Web, and shown to perform remarkably on a variety of downstream applications (*e.g.*, Tan and Bansal, 2019; Bugliarello et al., 2021; Radford et al., 2021; Li et al., 2021; Zeng et al., 2022; Gan et al., 2022). Nonetheless, recent work has highlighted their limitations in *fine-grained* tasks, where precise understanding of both modalities is required to correctly select a positive match against a negative one. Examples of such tasks include verb understanding (*sitting* vs. *standing*; Hendricks and Nematzadeh 2021), word order (*water in bottle* vs. *bottle in water*; Thrush et al. 2022), spatial relations (*above* vs. *below*; Liu et al. 2023) and other linguistic phenomena (Parcalabescu et al., 2022; Nikolaus et al., 2022; Yuksekgonul et al., 2023).

---

[*]Work completed during an internship at DeepMind. [‡]denotes equal senior contribution. Correspondence to: Emanuele Bugliarello <emanuele@di.ku.dk>.

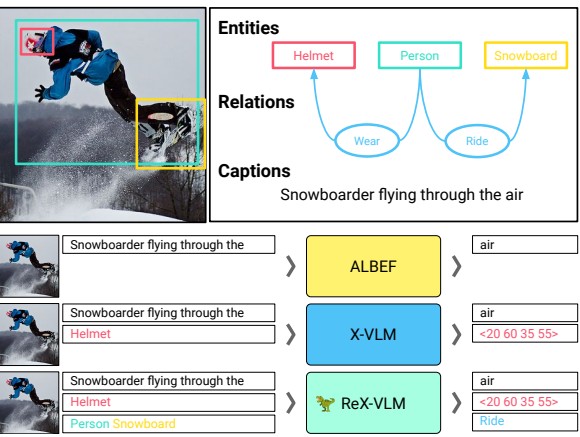

Figure 1: Overview of (i) the types of image annotations and (ii) the pretraining tasks and models used in this work. In addition to captions and entities, we study the benefits of modelling visual relations that link entities.

Recent work (Yao et al., 2022b; Zeng et al., 2022; Zhang et al., 2022, *i.a.*) shows that leveraging entity localisation, such as bounding boxes, from supervised data improves performance on downstream tasks like visual QA (Antol et al., 2015) and visual reasoning (Suhr et al., 2019). Interestingly, modelling visual locations is also crucial to learn fine-grained image–text mappings (Bugliarello et al., 2023). Motivated by this promising research thread, in this work, we further explore the benefits of supervised data in multimodal pretraining by leveraging structured visual information in the form of relations in scene graphs (Elliott and Keller, 2013; Johnson et al., 2015; Krishna et al., 2017).

A *scene graph* is a data structure that describes the content of a visual scene by expressing its entities (*e.g.*, *helmet*), their attributes (*e.g.*, *red*), and their relationships (*e.g.*, person *wear* red helmet; see Figure 1). While a large body of work has focused on generating scene graphs (Lu et al., 2016; Xu et al., 2017; Peyre et al., 2017; Zellers et al., 2018; Tang et al., 2020; Sharifzadeh et al., 2021, 2022; Chang et al., 2023, *inter alia*), they have also been used in other applications, such as im-

age retrieval (Johnson et al., 2015), image generation (Johnson et al., 2018) and image captioning (Yao et al., 2018; Yang et al., 2019).

However, the use of scene graphs in vision-and-language (V&L) pretraining has received limited attention. In contrast to prior work (Yu et al., 2021; Lee and Kang, 2022) that relied on relations inferred from captions through masked language modelling, we use a small dataset of human-annotated scene graphs, which go beyond the salient ones that are referred in a caption. To make full use of this rich learning signal, we introduce (i) a novel objective and (ii) a new method for data-to-text generation, which both explicitly aim to induce structure in the image and its connection in text. Our results show that modelling a limited amount of scene graph data in addition to millions of Web-crawled image–text pairs further improves coarse- and fine-grained skills in VLMs.

**Contributions**. In this work, **1)** we aim at improving fine-grained understanding in VLMs by modelling the *structure* of visual scenes during pretraining. In particular, we rely on a *small* amount of human-annotated scene graphs,[1] and propose two novel pretraining approaches: *verbalised scene graphs* (VSG) and *masked relation classification* (MRC). When compared to strong baselines, **2)** we show their effectiveness during pretraining on both fine- and coarse-grained zero-shot tasks. Our models achieve overall better fine-grained abilities, such as state-of-the-art in visual spatial reasoning, whilst keeping competitive performance on coarse-grained retrieval. **3)** We shed light on the individual contributions and interactions of our proposed methods. We find, for instance, that VSG enables dense caption understanding, and that, at scale, modelling relations can be more effective than modelling entity locations for fine-grained understanding. Finally, **4)** we revisit the standard practice of selecting the last checkpoint in V&L pretraining, showing that COCO Dev TR@1 leads to better models, especially on coarse-grained tasks.

## 2 Learning Visual Relations

Recent work in modelling spatial layout of entities in an image (Zeng et al., 2022) has been shown to be effective for fine-grained V&L understanding (Bugliarello et al., 2023). Yet, the information

of a visual scene goes beyond individual entities, and understanding their semantic relationships is a key step towards better VLMs. Nevertheless, this is an under-explored area of research. We hence investigate two approaches to better impose the structure of visual scenes from scene graphs (see Figure 2). The first approach, *verbalised scene graphs* (VSG), provides a different view of an image by associating it with a description of the relationships between entities in the image. Our second approach, *masked relation classification* (MRC), predicts the relation between two entities in the same image when their cross-modal representations are obtained from visually masked contexts.

**Setup.** Given an image $\mathcal{I}$, it can be associated with three types of annotations in our framework. $\mathcal{C}_{\mathcal{I}} = \{c_i\}$ denotes a collection of strings that describe the image $\mathcal{I}$ (*i.e.*, captions). $\mathcal{E}_{\mathcal{I}} = \{\mathbf{e}_i\}$ is a set of entities present in the image. Entities are defined by a label $l_i$ (*i.e.*, a string such as "cat" or "duck with sunglasses") and their spatial location—as bounding box coordinates—in the image: $\mathbf{e}_i = (l_i, x_i^{\min}, y_i^{\min}, x_i^{\max}, y_i^{\max})$. $\mathcal{G}_{\mathcal{I}} = \{\langle \mathbf{e}^s, r, \mathbf{e}^o \rangle_i\}$ is a collection of subject–relation–object triplets linking two entities ($\mathbf{e}_i^s$ and $\mathbf{e}_i^o$) via a string ($r_i$) of their spatial relation, such as "below" or "in front of."

### 2.1 VSG: Verbalised Scene Graphs

Inspired by Bugliarello and Elliott (2021); Yu et al. (2021) and work in data-to-text generation (*e.g.*, Kukich, 1983; Gardent et al., 2017; Agarwal et al., 2021), we explore the role of scene graph annotations for fine-grained understanding by generating text descriptions that encode entities and relations.

Given an image $\mathcal{I}$ and its scene graph $\mathcal{G}_{\mathcal{I}}$, we first `sample` $K$ triplets from $\mathcal{G}_{\mathcal{I}}$, $\langle \mathbf{g}_1, \ldots, \mathbf{g}_K \rangle$. Second, we ensure a fixed order in the triplets by `sorting` them based on the spatial location of their subject entities, represented by their centre location, $\langle \mathbf{g}_{\bar{1}}, \ldots, \mathbf{g}_{\bar{K}} \rangle$. Finally, we `verbalise` them into a single caption: "[CLS] $l_{\bar{1}}^s \, r_{\bar{1}} \, l_{\bar{1}}^o$ [SEP] $\ldots \, l_{\bar{K}}^s \, r_{\bar{K}} \, l_{\bar{K}}^o$ [SEP]," where [CLS] and [SEP] are special text tokens used in our baselines to learn a sentence-level representation and to separate between two phrases, respectively.

As shown in Figure 2 (left), once verbalised, the resulting scene graph strings are simply treated analogously to image captions $\mathcal{C}_{\mathcal{I}}$. In our experiments, our models are pretrained with the three objectives used by ALBEF (Li et al., 2021): $\mathcal{L}_A = \mathcal{L}_{CL} + \mathcal{L}_{ITM} + \mathcal{L}_{MLM}$; where $\mathcal{L}_{MLM}$ is the masked

---

[1] We note that 'learning where only a subset of training data is given with labels' (*i.e.*, incomplete supervision) is one of three types of weak supervision. We refer to Zhou (2017) for a relevant review of research in weakly-supervised learning.

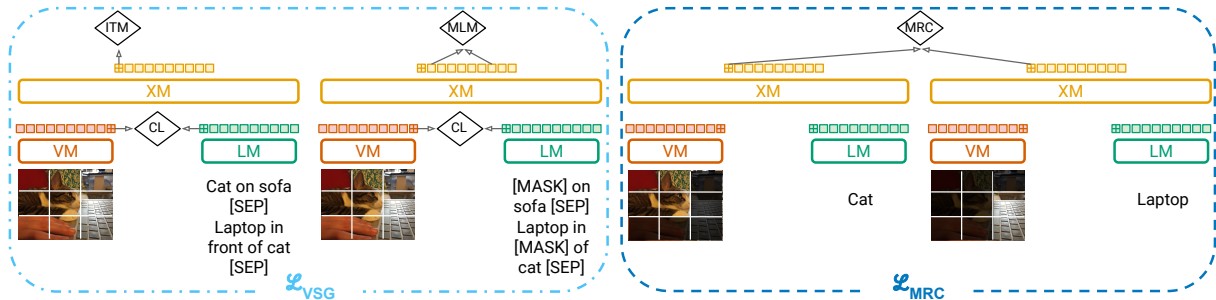

Figure 2: Overview of our proposed approaches. VSG applies the standard pretraining objectives used on image captions to verbalised scene graph annotations. MRC predicts the relation (*e.g.*, "in front of") from a predefined vocabulary between two entities when encoded with their visual context masked (shown as dark patches here). VM: vision model; LM: language model; XM: cross-modal model; CL: contrastive learning loss; ITM: image–text matching loss; MLM: masked language modelling loss; MRC: masked relation classification loss.

language modelling loss, $\mathcal{L}_{CL}$ is the image–text contrastive learning loss, and $\mathcal{L}_{ITM}$ is the cross-modal image–text matching loss. That is, $\mathcal{L}_{VSG}$ is equivalent to $\mathcal{L}_A$ but applied to verbalised scene graphs; but note that VSG data could, in theory, be used with any image–text losses applied in VLMs.

## 2.2 MRC: Masked Relation Classification

Our second proposed objective aims at following the progress that masked predictions have had in NLP (*e.g.*, Devlin et al., 2019; Zhang et al., 2019) and Computer Vision (*e.g.*, Bao et al., 2022; He et al., 2022). In particular, we were inspired by X-VLM (Zeng et al., 2022), which learns to better localise entities by solely considering an entity's image region when applying image–text losses.

As shown in Figure 2 (right), given a scene graph triplet $\langle \mathbf{e}^s, r, \mathbf{e}^o \rangle$ sampled from $\mathcal{G}_\mathcal{I}$, we first separately encode its subject and object entities by masking their visual context. Second, we `pool` the final cross-modal representation for the two entities (represented by the final features of the `[CLS]` token in our models). Finally, we `concatenate` them into a single vector, which is then processed by a two-layer MLP and mapped to an output space of $V$ labels, corresponding to the top-$V$ most frequent relations in our scene graph data. The model is then trained to predict (*i.e.*, classify) the correct subject–object relation with a cross-entropy loss.

## 3 Experimental Setup

We validate the effectiveness of our approaches by enhancing two strong VLMs on four, diverse fine-grained and two coarse-grained benchmarks. App. A provides details to reproduce our work. Our models can be accessed and verified online.[2]

**Models.** We focus our analysis on four models: ALBEF and X-VLM, and their corresponding relation-enhanced models (REALBEF and REX-VLM, respectively). For reference, we also test two strong systems: CLIP (Radford et al., 2021), a popular dual-encoder; and BLIP-2 (Li et al., 2023), a VLM with frozen large image and text models.

**ALBEF** (Li et al., 2021) is a widely used VLM that achieves strong downstream performance by effectively combining key components for V&L learning, such as a contrastive objective and cross-attention, in its design. In ALBEF, an image and a caption are first independently encoded with a vision (ViT; Dosovitskiy et al. 2021; Touvron et al. 2021) and a text (BERT; Devlin et al. 2019) Transformer (Vaswani et al., 2017), respectively; and then fused in a dual-stream cross-modal Transformer (Bugliarello et al., 2021). The model is pretrained with three objectives (*cf.*, Section 2.1): masked language modelling, image–text contrastive learning, and image–text matching.

**X-VLM** (Zeng et al., 2022) uses the same components and objectives as ALBEF, but additionally learns to locate visual concepts in the image given the associated texts. It does so by predicting an entity's bounding box (bbox) coordinates given the visually grounded representation of its label (*e.g.*, 'black swan'). Moreover, Bugliarello et al. (2023) showed that X-VLM also learns to ground an object label by applying ALBEF's losses to a visually-masked image, which they collectively refer to as the visually-masked ALBEF (VMA) loss. These objectives allow it to acquire strong fine-grained understanding abilities, outperforming larger models, such as Flamingo (Alayrac et al., 2022) and BLIP-2 (Li et al., 2023), on these tasks.

---

[2]https://github.com/e-bug/weak-relation-vlm.

| Dataset | # Img | # Cap | # Ann |
|---|---|---|---|
| *Image captions* | | | |
| SBU (Ordonez et al., 2011) | 0.9M | 0.9M | - |
| COCO (Lin et al., 2014) | 0.1M | 0.5M | - |
| VG (Krishna et al., 2017) | 0.1M | 0.8M | - |
| $CC_{3M}$ (Sharma et al., 2018) | 1.8M | 1.8M | - |
| $CC_{12M}$ (Changpinyo et al., 2021) | 11.2M | 11.2M | - |
| *Object detection* | | | |
| $COCO_{OD}$ (Lin et al., 2014) | 0.1M | - | 0.4M |
| $VG_{OD}$ (Krishna et al., 2017) | 0.1M | - | 1.8M |
| $VG_{RD}$ (Krishna et al., 2017) | 0.1M | - | 2.9M |
| *Scene graphs* | | | |
| GQA (Hudson and Manning, 2019) | 0.1M | - | 1.4M |

Table 1: Statistics of our pretraining corpora.

**Pretraining data.** We pretrain all models for 200K/500K steps on the *same*, publicly available 4M/14M datasets originally used by the authors,[3] and, unless otherwise specified, we use the final checkpoint for evaluation. In particular, we rely on three types of pretraining data: image captions, object detection and scene graphs. Table 1 lists their statistics, where '# Ann' denotes the total number of entities identified by bbox–label pairs in object detection data, and the total number of relations in scene graphs. The unique number of relation strings in GQA scene graphs (expanding the original ones in VG) is equal to 310, which determines the size of the output vocabulary for our masked relation classification (MRC) method.

**Benchmarks.** We report zero-shot performance on coarse-grained retrieval in **Flickr30K** (Young et al., 2014) and **COCO** (Lin et al., 2014), and on four English fine-grained understanding datasets.

**VSR** (Liu et al., 2023) tests for 65 types of visual spatial relationships (*e.g.*, under, in front of) grouped into seven categories (*e.g.*, adjacency, orientation). Each sample consists of an image–sentence pair; a model needs to predict whether the sentence correctly describes the spatial relation between two entities in the image. We zero-shot evaluate models on the 'random' split, and report accuracy on both the Dev and Test sets due to their low correlation (Bugliarello et al., 2023).

**VALSE** (Parcalabescu et al., 2022) examines six linguistic phenomena, such as plurality, actions and coreference. Given an image, a model is asked to distinguish real captions from foils (Shekhar et al., 2017), where a foil is constructed from a caption by altering a word or phrase that realises a specific linguistic phenomenon (*e.g.*, saying that an image shows *six* zebras instead of *four* for counting).

**SVO-Probes** (Hendricks and Nematzadeh, 2021) evaluates verb understanding by asking a model to compare a caption with two images: one that matches it, and one that is semantically different in its corresponding subject, verb, or object.

**Stanford Paragraphs** (Krause et al., 2017) is a dataset of paragraphs describing images in unified stories (one paragraph annotation per image). Paragraphs give a coherent natural language description for images, requiring both fine-grained image understanding and long-term language reasoning.[4]

All these tasks are framed as image–text matching, a common pretraining objective of VLMs. On VSR, a model's prediction is correct if the matching score is greater/lower than 50% for a true/false label. On the other benchmarks, a model's prediction is correct if the score for the positive image–text pair is higher than that of the negative pair(s). Moreover, by evaluating through foils, which contain a single difference compared to the truth (*e.g.*, only a word differs between true and foil captions), VALSE and SVO-Probes allow to quantitatively measure specific fine-grained V&L abilities.

## 4 Results

Table 2 shows performance on fine-grained tasks that cover a wide range of multimodal abilities of our baselines, relation-enhanced models, as well as current strong dual- and cross-encoder models.[5]

**Enhanced visual spatial reasoning capabilities.** We validate the effectiveness of our proposed approaches in modelling visual relations by evaluating on the task of visual spatial reasoning (VSR). This is a benchmark that focuses on spatial relations, and we expect our approaches to significantly improve upon their baselines here. Our proposed REX-VLM$_{13M}$ model substantially outperforms its X-VLM$_{13M}$ baseline by +6.8/3.0pp on the Dev/Test sets, setting a new state-of-the-art on zero-shot VSR. Moreover, REX-VLM$_{13M}$ consistently outperforms the other models on the related subtasks of 'spatial relations' and 'actant swap' of VALSE (see Table 5 in App. B.1). We also observe

---

[3]We note that only 1.8M and 11.2M data points were available for $CC_{3M}$ and $CC_{12M}$, respectively, at our time of study.

[4]Note that the images in Stanford Paragraphs are a subset of VG. Recent VLMs use VG data during pretraining, which could justify the high performance we observe in our results.

[5]Though these fine-grained tasks do not explicitly require scene graph understanding or generation, we hypothesise that by training with this data, our models will gain better fine-grained image–text understanding.

Table 2:

| Model | | VSR Random | VALSE | SVO-Probes | Stanford Paragraphs | |
| Name | Role | Dev / Test Acc | $Acc_r$ | $Acc_r$ | IR@1/5 | TR@1/5 |
|---|---|---|---|---|---|---|
| $CLIP_{400M}$ | | N/A[†] | 64.0 | 81.6 | 45.3 / 73.1 | 53.4 / 80.1 |
| $BLIP-2_{129M}$ | | 61.2 / 61.5 | 74.0 | 86.5 | 83.4 / 95.2 | 81.1 / 94.3 |
| $ALBEF_{3M}$ | BASELINE | 63.7 / 60.1 | 69.4 | 86.6 | 79.5 / 95.6 | 79.8 / 94.9 |
| $REALBEF_{3M}$ | +RELATIONS | 64.0 / 60.2 | 69.6 | 86.2 | 85.1 / 97.4 | 85.8 / 97.2 |
| $X-VLM_{3M}$ | +LOCALISATION | 63.5 / **62.3** | 69.5 | **87.3** | 79.8 / 94.8 | 81.4 / 95.0 |
| $REX-VLM_{3M}$ | +BOTH | **65.0** / 61.8 | **70.9** | **87.3** | **87.4** / **97.8** | **87.8** / **97.4** |
| $ALBEF_{13M}$ | BASELINE | 60.4 / 59.4 | 72.2 | 86.7 | 77.1 / 93.7 | 73.7 / 90.3 |
| $REALBEF_{13M}$ | +RELATIONS | 64.6 / 61.3 | 70.4 | 87.5 | 86.7 / 97.5 | 86.5 / 97.2 |
| $X-VLM_{13M}$ | +LOCALISATION | 61.1 / 60.5 | 71.3 | 87.3 | 80.3 / 94.9 | 76.8 / 92.4 |
| $REX-VLM_{13M}$ | +BOTH | **68.4** / **63.5** | **73.3** | **88.1** | **89.3** / **98.0** | **88.8** / **97.7** |

Table 2: Overall results on zero-shot fine-grained benchmarks. Models pretrained on 3M/13M images are evaluated after 200K/500K steps, respectively. Values underlined in green (red) denote gains (losses) of relation-enhanced models on their baselines. [†]CLIP cannot be directly evaluated on VSR since it requires true/false predictions for a given image–text input, while CLIP is only trained with a contrastive loss. Best results are in **bold**.

consistent gains when modelling relations on top of ALBEF (REALBEF$_{13M}$ gains +3.8/0.8pp), which shows that even just modelling relations (without modelling objects) is helpful for VSR. These results show that our approaches to modelling relations play an important role in tasks that require spatial reasoning. Finally, we see that modelling visual relations when pretraining on fewer images only results in slightly better VSR Dev accuracy. It is interesting to note that both our models further increase their performance gains when moving from 3M to 13M images, despite now only having 0.6% of the images annotated with scene graphs.

**Improved fine-grained understanding.** In addition to VSR, REX-VLM$_{13M}$ performs best across all the fine-grained tasks, which test models for a much broader range of fine-grained skills. It gains +1.7pp on VALSE and +0.8pp on SVO-Probes, and REX-VLM$_{3M}$ gains +1.4pp on VALSE. These results confirm that visual relations can provide useful signal towards fine-grained understanding, even when only available for a tiny percentage of pretraining data. On the other hand, REALBEF models are on par with their baselines. Recall that they model relations between entities without explicitly learning about the entities themselves. That is, it is harder for ALBEF models to learn relations without doing localisation. Moreover, comparing X-VLM and REALBEF on VALSE and SVO-Probes, we see that modelling objects (X-VLM) on top of ALBEF is slightly better than solely modelling relations (REALBEF).

**Substantially better fine-grained understanding on dense captions.** Thrush et al. (2022) showed that current VLMs struggle more when matching

| Model | Flickr30K | | COCO | |
| | IR@1/5 | TR@1/5 | IR@1/5 | TR@1/5 |
|---|---|---|---|---|
| $CLIP_{400M}$ | 88.0 / 98.7 | 68.7 / 90.6 | 58.4 / 81.5 | 37.8 / 62.4 |
| $BLIP-2_{129M}$ | 95.5 / 99.9 | 86.7 / 97.1 | 80.7 / 94.7 | 64.2 / 85.2 |
| $ALBEF_{3M}$ | 77.9 / 92.7 | 61.3 / 83.6 | 63.6 / 86.1 | 47.4 / 74.5 |
| $REALBEF_{3M}$ | 75.5 / 92.3 | 59.5 / 82.6 | 62.7 / 86.2 | 46.4 / 74.7 |
| $X-VLM_{3M}$ | 78.2 / **94.1** | **61.8** / 84.1 | 63.9 / 86.7 | 47.9 / 74.7 |
| $REX-VLM_{3M}$ | **78.6** / 93.9 | **61.8** / **84.8** | **66.0** / **87.5** | **48.9** / **76.2** |
| $ALBEF_{13M}$ | **82.2** / 95.5 | 66.1 / 85.8 | 64.8 / 86.6 | 49.1 / 74.6 |
| $REALBEF_{13M}$ | 80.3 / 93.7 | 65.3 / 86.0 | 66.4 / 87.8 | 49.1 / 76.1 |
| $X-VLM_{13M}$ | 83.3 / 95.6 | 66.2 / 86.3 | 64.1 / 86.4 | 49.1 / 74.8 |
| $REX-VLM_{13M}$ | 80.3 / **95.8** | **66.6** / **87.0** | **66.9** / **88.9** | **50.2** / **77.0** |

Table 3: Overall results on zero-shot coarse-grained benchmarks. Models pretrained on 3M/13M images are evaluated after 200K/500K steps, respectively. Values in green (red) denote gains (losses) of relation-enhanced models on their baselines. Best results are in **bold**.

captions with two main predicates than one. We thus consider testing our models for the ability to understand long, fine-grained descriptions of images on the task of zero-shot image–paragraph retrieval. In fact, paragraphs are longer, more informative, and more linguistically complex than sentence-level captions. REX-VLM$_{13M}$ achieves 89.3 TR@1 and 88.8 IR@1 (+9.0pp and +12.0pp compared to X-VLM$_{13M}$). Such high performance is largely preserved when training on 3M images (87.4pp and 87.8pp), and it carries over to REALBEF models as well. Overall, relation-enhanced models gain from +5.6pp to +12.8pp on this task.

**Competitive coarse-grained retrieval.** Finally, we evaluate our relation-enhanced models on zero-shot image–text retrieval tasks to verify that their gains on fine-grained tasks do not hamper performance on coarse-grained tasks. Table 3 lists performance on the Flickr30K and COCO datasets. Our REX-VLM models achieve similar or better per-

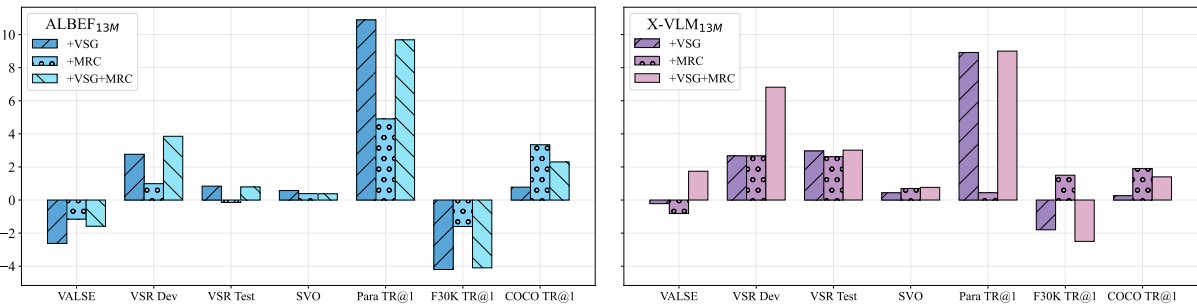

Figure 3: Difference in performance when adding our approaches to ALBEF$_{13M}$ and X-VLM$_{13M}$ models.

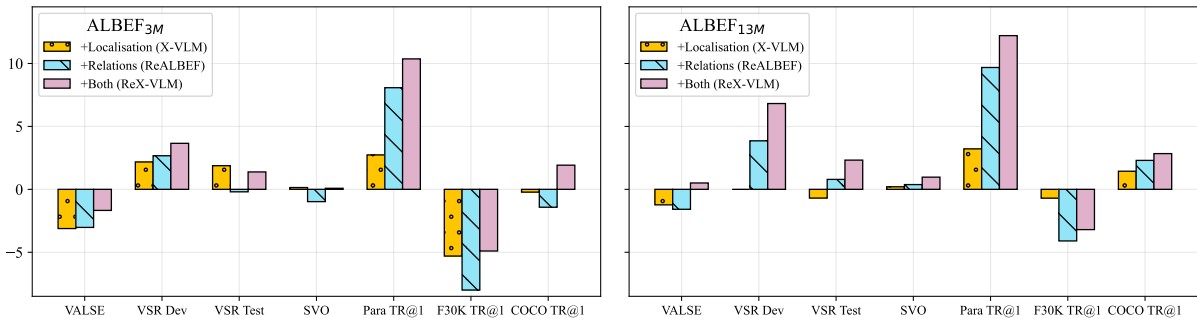

Figure 4: Performance difference when adding supervision for localisation, relations and both approaches.

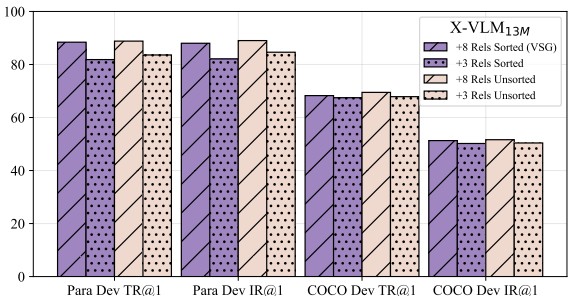

Figure 5: Ablations of our VSG approach.

formance than their X-VLM baselines, especially on COCO (+2.8pp TR@1 and +1.2 IR@1). That is, learning visual relations from a small amount of annotated images is especially effective on in-domain data (relation-annotated images are a subset of COCO images) for the task of zero-shot image–text retrieval. On the other hand, REALBEF models tend to perform slightly worse than their baselines, especially on Flickr30K and when trained for 200K steps on 3M images. This shows that modelling relations without objects hinders performance on coarse-grained understanding in this setup. Figures 8 and 9 (App. B.2) show that this is due to sub-optimal data mixing rather than modelling limitations at scale. For instance, REALBEF$_{3M}$ matches or outperforms ALBEF$_{3M}$ when trained for longer.

**Approach ablations.** Figure 3 shows the individual contributions of our proposed approaches towards the final models' performance. Looking at our REX-VLM models, we see that combin-

ing both VSG and MRC typically leads to the best performance. On VALSE, we find that using either approach independently decreases accuracy, while using them together increases it. It is clear that VSG is instrumental to perform well on image–paragraph retrieval for both models. However, VSG hurts performance on coarse-grained retrieval tasks. This is likely because scene graphs are treated equally to image captions here, although being distributionally different. Finally, we see similar patterns for ALBEF models, although they often gain more from VSG than from MRC.

**What matters for long context understanding?** As discussed above, our VSG approach plays a crucial role towards dense caption understanding. In VSG, we propose an alternative view of an image by creating a textual description that consists of a sequence of subject–relation–object triplets sampled from the image's scene graph. In our main approach, we verbalise scene graph annotations by (i) sampling 8 relations per image, and (ii) sorting them based on the subject entity's bounding box coordinates. We ablate both of these choices in Figure 5, where we pretrained X-VLM$_{13M}$ models for 200K steps by adding VSG variants with (i) fewer relations (3 instead of 8), and (ii) without sorting them. We find that while sorting the relations is not critical to perform well on Stanford Paragraphs, the number of relations is an important factor. This not only significantly boosts image–paragraph re-

trieval, but also leads to smaller yet consistent gains on COCO, which contrasts previous findings on the effectiveness of long descriptions for caption retrieval (Hendricks et al., 2021). We hypothesise this is due to the nature of our descriptions, which encode local relations between entities, rather than being long, complex natural captions.

**Localisation or relations?** We finally analyse whether it is more effective to model visual locations of entities or their relations in visual scenes. To do so, we compare the performance gains obtained on top of ALBEF by X-VLM (localisation), REALBEF (relations) and REX-VLM (both). For models trained on 13M images, Figure 4 (right) shows that modelling relations alone is typically *better* than modelling objects. A notable exception is coarse-grained retrieval on out-of-distribution Flickr30K, where both modelling localisation and, especially, relations decrease performance. Combining both objectives results in the best results. When training on 3M images, Figure 4 (left) shows similar results but with object localisation giving larger contributions. Taken together, we see that current VLMs can learn more from modelling relations than localisation when trained at scale. Finally, we see that adding and modelling a small amount of supervised data (as done by REALBEF$_{3M}$, X-VLM$_{3M}$ and REX-VLM$_{3M}$) is typically more effective than adding 11M additional image–text pairs crawled from the Web (*i.e.*, ALBEF$_{13M}$) for fine-grained understanding.

## 5  Analysis of Learning Dynamics

Recent work on V&L pretraining typically trains for a fixed number of steps, and then selects the last checkpoint to report performance on different tasks. However, Bugliarello et al. (2023) showed that current, strong models achieve peak performance on different fine-grained tasks at different stages of pretraining. This motivates us to study the pretraining dynamics of our models, and to reassess the current practice of choosing the last checkpoint by investigating how different checkpoint selection strategies affect the performance on our tasks.

**Convergence rates.** Performance for X-VLM models pretrained on 13M images is shown in Figure 6.[6] We find that our REX-VLM model requires longer training to achieve peak performance across

fine-grained tasks. In fact, while our baseline's performance starts degrading after 250K steps, REX-VLM continues improving over time, reaching its best yet results at 500K steps.[7] We can also see that, by the end of our training, relation-enhanced models typically achieve better performance than the best results given by our baselines, confirming the validity of our results from the previous section. Likewise, the evaluation curves show that our models and baselines can achieve comparable coarse-grained retrieval performance, and that longer training can help relation-enhanced models close the gap with the baselines (see Figure 8 in App. B.2). Given a fixed number of steps, we leave it for future work to investigate pretraining schedules that better balance coarse- and fine-grained tasks so as to obtain a single checkpoint that performs well on both kinds of skills.

**Checkpoint selection strategies.** As shown above, our relation-enhanced models achieve highest performance on most tasks at the end of training. On the other hand, this is not the case for our ALBEF and X-VLM baselines. We hence revisit the concept of checkpoint selection in pretrained VLMs, as recent work simply trains for a fixed number of steps (*e.g.*, 200K or 500K). Specifically, we analyse how using different tasks (Dev split when possible) for checkpoint selection affects performance on other benchmarks. That is, for each model, we select the checkpoint that gives the highest score on a given Dev task, and evaluate it across tasks. In Figure 7, we show the difference in performance ($y$-axis) obtained using different checkpoint selection strategies ($x$-axis) compared to the fixed checkpoint results reported in Tables 2 and 3, averaged across all models.[8] Overall, we find that COCO Dev TR@1 leads to better checkpoint selection for all coarse-grained benchmarks (and a small improvement overall). However, we do not see a consistent pattern for fine-grained tasks, probably because they are more varied in terms of skills they test compared to coarse-grained retrieval tasks. For instance, using SVO-Probes results in better VSR but worse VALSE performance. Table 8 in App. B shows fine- and coarse-grained performance when selecting checkpoints based on COCO Dev TR@1. While REX-VLM still outper-

---

[6]Figures 8 and 9 in App. B.2 show similar patterns for all the models pretrained on 3M and 13M images, respectively.

[7]We note that fine-grained understanding of REALBEF$_{3M}$ and REX-VLM$_{3M}$ start degrading after 350K steps (App. B).

[8]In Figure 7, our average results do not include REALBEF$_{13M}$ and REX-VLM$_{13M}$ as they consistently achieved highest performance at the last, fixed checkpoint.

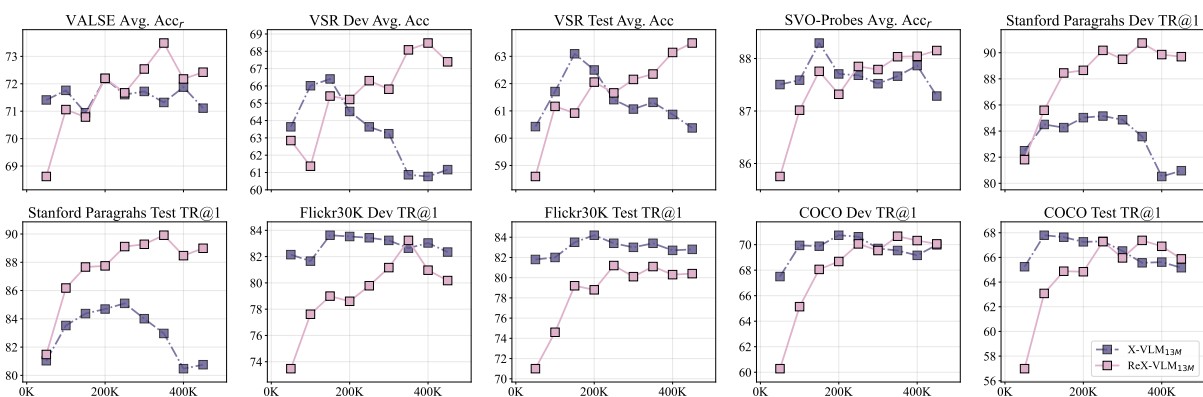

Figure 6: Pretraining dynamics of our X-VLM models when learning from 13M images.

| | F30K Dev IR@1 | F30K Dev TR@1 | COCO Dev IR@1 | COCO Dev TR@1 | VALSE Avg. Acc$_r$ | SVO Avg. Acc$_r$ |
|---|---|---|---|---|---|---|
| F30K Test IR@1 | 0.87 | 0.15 | 0.79 | 1.10 | 0.44 | 0.28 |
| F30K Test TR@1 | 0.67 | -0.23 | 0.85 | 0.78 | 0.02 | 0.12 |
| COCO Test IR@1 | 0.65 | 0.26 | 0.55 | 0.75 | 0.20 | 0.33 |
| COCO Test TR@1 | 1.36 | 0.59 | 1.17 | 1.57 | 0.36 | 0.66 |
| VALSE Avg. Acc$_r$ | -0.29 | -0.06 | -0.26 | -0.04 | 0.54 | -0.46 |
| SVO Avg. Acc$_r$ | 0.33 | 0.15 | 0.12 | 0.31 | -0.05 | 0.53 |
| VSR Dev Avg. Acc | 0.63 | 0.77 | 0.49 | 0.89 | 0.31 | 0.97 |
| VSR Test Avg. Acc | 0.34 | 0.67 | 0.68 | 0.68 | -0.18 | 0.85 |
| *Cross-task Avg.* | *0.57* | *0.29* | *0.55* | ***0.76*** | *0.20* | *0.41* |

Figure 7: Average test performance differences ($y$-axis) with respect to fixed checkpoints for all models except for REALBEF$_{13M}$ and REX-VLM$_{13M}$ according to different checkpoint selection tasks ($x$-axis). For detailed results on each model, see Figure 10 in App. B.3.

forms the other models on fine-grained tasks, we find that the baselines perform on par on coarse-grained tasks. Finally, we note that, while different checkpoint selection strategies result in slight variations in task performance, the ranking of models does not change much. This is shown in Figure 11 (App. B.3), where we compute the Spearman rank correlation coefficients between COCO Dev TR@1 and the other strategies, across all models. The high rank correlation coefficients across all strategies and evaluation tasks demonstrates that REX-VLM robustly outperforms the other models.

## 6 Related Work

**Fine-grained VLMs.** While the vast majority of VLMs are solely pretrained on large-scale data collected from the Web (*e.g.*, Lu et al., 2019; Chen et al., 2020; Radford et al., 2021; Alayrac et al., 2022; Yu et al., 2022; Li et al., 2022b, 2023), a recent line of work investigates the challenge of learning fine-grained image–text mappings. FILIP (Yao et al., 2022a), LOUPE (Li et al., 2022a), RegionCLIP (Zhong et al., 2022), PyramidCLIP (Gao et al., 2022) and HiCLIP (Geng et al., 2023) propose different fine-grained alignment methods for dual-encoder networks. On the other hand, GLIP (Li et al., 2022c; Zhang et al., 2022), Fiber (Dou et al., 2022), PEVL (Yao et al., 2022b), MVPTR (Li et al., 2022d), X-VLM (Zeng et al., 2022) and PaLI (Chen et al., 2023b) show the benefits of learning cross-modal representations from additional supervised object detection data. Finally, there is increasing interest in training VLMs that perform well on a range of coarse- and fine-grained vision and language tasks (Lu et al., 2020; Wang et al., 2022; Lu et al., 2023; Zou et al., 2023; Chen et al., 2023a; Beyer et al., 2023).

**Scene graphs and multimodality.** The structural representations of scene graphs has been explored in the context of different V&L tasks, such as image–text retrieval (Johnson et al., 2015; Schuster et al., 2015; Schroeder and Tripathi, 2020; Ge et al., 2023), image captioning (Yao et al., 2018; Yang et al., 2019), and visual QA (Qian et al., 2022; Koner et al., 2021; Lee et al., 2019; Shi et al., 2019). Only two studies have, however, investigated the role of scene graphs in V&L pretraining. ERNIE-ViL (Yu et al., 2021) first extracts scene graphs from the captions with an off-the-shelf model, and then proposes MLM-based object, attribute, and relationship prediction tasks to learn cross-modal detailed semantic alignments. Lee and Kang (2022), in addition to extracting subject–relation–object triplets from captions with an off-the-shelf model,

also generate paired visual features based on the entities output by an object detection model and their co-occurrences in the VG dataset (Krishna et al., 2017). Unlike them, we rely on a *small* sample of human-annotated scene graphs, and propose two methods for relation prediction in V&L pretraining. Furthermore, we are the first to show the benefits of modelling scene graphs towards acquiring better fine-grained skills during multimodal pretraining.

# 7  Conclusion

Previous work in multimodal pretraining has shown the importance of modelling objects (using localisation data) in improving the performance of both coarse- and fine-grained tasks. In this paper, we investigate if supervision from relational data—by modelling relations between objects in a visual scene—can improve performance on these tasks. In particular, we rely on scene graph annotations, an under-explored data structure for multimodal pretraining, and propose two approaches for leveraging relations between entities in an image: 1) MRC, a pretraining objective that predicts the relation between the objects in two image regions; and 2) VSG, a versatile data-to-text generation recipe that converts scene graphs into captions, that can then be fed to any VLM. When applied to strong VLMs, we find that our methods improve their fine-grained understanding, with REX-VLM achieving state-of-the-art spatial reasoning abilities, as well as strong performance on other tasks too.

We hope that our work motivates further research in improving fine-grained understanding in VLMs. Given the promise of our results with a few annotated images, an interesting future direction is to study how to best scale up our approaches with machine generated data, *e.g.*, by generating pseudo-labels from off-the-shelf scene graph generators from either images or captions, or both.

## Limitations

Our paper investigates the benefits and limitations of learning structured information in visual scenes from scene graph annotations.

Collecting such rich annotations from humans is time consuming, and it cannot be easily scaled up to millions of images. While our work shows that models pretrained at scale can still benefit from a limited number of scene graphs, differences were less significant on out-of-distribution images. This aspect is especially relevant in a multilingual setup—wherein the data can contain concepts beyond those represented in English and Western societies (Liu et al., 2021)—and towards safe and reliable deployment of multimodal systems. A promising direction to mitigate this limitation is to devise bootstrapping strategies to enrich a massive number of images with rich scene graph annotations.

From an experimental angle, we measure zero-shot performance of pretrained vision-and-language models (VLMs). Due to resource constraints, we only pretrain our models once. Although we observe consistent gains of our approaches with respect to their baselines, we note that Bugliarello et al. (2021) showed that pretraining a given model with different seeds can result in different performance when fine-tuned on several downstream tasks, like visual question answering or visually grounded reasoning. Further investigation is required to assess the variance of pretrained VLMs in zero-shot (fine-grained) evaluations.

Moreover, even though the proposed approaches can be applied to most recent VLMs, we only evaluate two architectures—ALBEF and X-VLM—due to computational constraints. Although X-VLM is the current state-of-the-art for most fine-grained understanding tasks, it would be instructive to measure how our approaches transfer to models that process images through learnable visual queries (Alayrac et al., 2022; Li et al., 2023, *i.a.*).

We also note that some of our evaluation datasets are quite small, and encourage the community to create larger evaluation sets to reliably measure progress in coarse- and fine-grained V&L skills.

Finally, in this paper, we revisit the idea of checkpoint selection for pretrained VLMs. While recent work simply trains for a fixed number of steps, we find that using COCO validation TR@1 leads to overall better models in our evaluations. Yet, our findings are only based on a handful of models. We encourage the community to investigate this line further, especially since current VLMs may learn different skills at different stages of pretraining.

## Ethics Statement

In this work, we include additional supervision to guide models into learning visual relations and improve performance on a variety of vision-and-language tasks. However, biases in multimodal datasets are well documented (Meister et al., 2022) and, without further mitigation, we expect our models to learn them. Furthermore, our datasets include

images with faces, and there is no mechanism for people to remove themselves from these datasets.

Multimodal models like ALBEF and X-VLM can be used for a variety of vision-and-language tasks including image and video retrieval, video description, and visual question answering. Beneficial applications of such models include better human–computer interaction, or visual description and question answering for the visually impaired. However, these models can also be used for harmful applications such as surveillance.

## Acknowledgements

The authors would like to thank Aishwarya Agrawal, Laurent Sartran, Jovana Mitrovic, Sahand Sharifzadeh, Chris Dyer and the Google DeepMind Language Team for feedback on this project.

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

## A  Experimental Setup

In this section, we provide further details on the experimental setups that we used for our studies.

Our ALBEF and X-VLM models are implemented in JAX (Babuschkin et al., 2020) and employ a ViT-B/16 image encoder pretrained on ImageNet-21k (Steiner et al., 2022) that processes images with a resolution of $224\times224$ pixels. AL-BEF models have 212M parameters, while X-VLM models have 214M parameters. For MRC, we use a two-layer MLP with a ReLU nonlinear activation function, further adding 5.7M parameters during pretraining. VSG is parameter-free.

We pretrain our baselines and relation-enhanced models on a $2\times2\times2$ TPUv4 slice for up to 500K steps (5 days). Each model is pretrained *once*, using the same hyperparameters as the baselines whenever applicable. For VSG, we sample 16 relations per image in order to fit within the TPU memory. For MRC, we follow the same setup as for VMA/BBOX in X-VLM, by sampling 4 entities per image (and their 2 corresponding relations). During pretraining, we group datasets according to their 'type' (*i.e.*, captions, detection or graphs), and sample batches containing data from a single dataset at a time. Within a group, we sample datasets uniformly at random, as this was shown to be more effective for captioning data (Hendricks et al., 2021). We also experimented with sampling VSG and MRC data with a weight of 1.5, but found 1.0 to lead to lower pretraining loss. We use a maximum sequence length of 36 text tokens for all tasks except for VSG, for which we use 112 tokens to fit up to 16 subject–relation–object triplets per caption. For masked language modelling tasks, we mask 25% of the text tokens in a caption, ensuring that all tokens that belong to a word are masked. Hyperparameter configurations for best-performing models are listed in Table 4.

We typically report performance after (i) 200K steps when training on 3M images, and (ii) 500K steps when training on 13M images. Compared to the total number of data points seen throughout pretraining in the original papers, our models are typically trained on fewer examples. Li et al. (2021) trained ALBEF $_{4M}$/ALBEF $_{14M}$ on 154.5/456M samples, while we use 102.5/256M samples to train ALBEF$_{3M}$/ALBEF$_{13M}$ and corresponding relation-enhanced models. Zeng et al. (2022) trained X-VLM $_{4M}$/X-VLM $_{16M}$ on approximately 315/921.5M samples, while we use 205/512M sam-

| Hyperparameter | ALBEF | ReALBEF | X-VLM | ReX-VLM |
|---|---|---|---|---|
| Learning rate | 1e-4 | 1e-4 | 1e-4 | 1e-4 |
| AdamW $\beta$ | (0.9, 0.995) | (0.9, 0.995) | (0.9, 0.95) | (0.9, 0.95) |
| Weight decay | 0.02 | 0.02 | 0.02 | 0.02 |
| Warmup steps | 5000 | 5000 | 5000 | 5000 |
| Dropout | 0.1 | 0.1 | 0.1 | 0.1 |
| # Entities (VMA/BBOX) | - | 4 | 4 | 4 |
| # Relations (MRC) | - | 2 | - | 2 |
| # Relations (VSG) | - | 8 | - | 8 |
| Batch sizes | 512:0:0:0 | 512:0:128:128 | 1024:1024:0:0 | 1024:1024:256:128 |
| Sampling ratios | 1:0:0:0 | 2:0:1:1 | 2:1.5:0:0 | 2:1.5:1:1 |

Table 4: Hyperparameter configurations for best-performing models. '# Entities (VMA/BBOX)' refers to the number of objects sampled from each image in a batch to compute the VMA and BBOX losses in X-VLM models. '# Relations (MRC)' refers to the number of subject–relation–object triplets sampled from each image in a batch to compute the MRC loss. '# Relations (VSG)' refers to the number of triplets sampled from each image in a batch to compute the VSG loss. Batch sizes and sampling ratios refer to different data types and losses, as captions:entities:MRC:VSG.

ples to train X-VLM$_{3M}$/X-VLM$_{13M}$ and corresponding relation-enhanced models.

## B  Results

In this section, we provide complementary results.

### B.1  Results by Subtask

Tables 5 to 7 list performance on the subtasks of our fine-grained benchmarks when pretraining our models for a fixed number of steps (see Section 4).

On VALSE, we find that REX-VLM models are especially useful to improve understanding of existence, counting (when pretrained on 3M images), spatial relations, actant swap and coreference. Their performance is on par in plurality, but note that the ALBEF$_{13M}$ baseline tops all other models on the coreference and Foil-it! subtasks.

On VSR, we observe significant, consistent gains of REX-VLM models in adjacency and projective relations. REX-VLM$_{13M}$ additionally boosts topological relations, while REX-VLM$_{3M}$ boosts directional relations. When learning relations on top of ALBEF, we observe similar trends for REALBEF$_{13M}$ but to a slightly smaller degree, indicating that it is helpful to learn object locations to better understand relationships between objects.

On SVO-Probes, REX-VLM$_{13M}$ gains +1pp on *subject* and *object* understanding, but less on verb understanding, compared to X-VLM$_{13M}$. The gains for subject understanding are even larger for REALBEF$_{13M}$ with respect to ALBEF$_{13M}$

| Model | Existence quantifiers | Plurality number | Counting | | | Sp.rel.‡ relations | Action | | Coreference | | Foil-it! | Avg. |
|---|---|---|---|---|---|---|---|---|---|---|---|---|
| | | | balanced | sns.† | adv.† | | repl.† | actant swap | standard | clean | | |
| Random | 50.0 | 50.0 | 50.0 | 50.0 | 50.0 | 50.0 | 50.0 | 50.0 | 50.0 | 50.0 | 50.0 | 50.0 |
| CLIP | 66.9 | 56.2 | 62.1 | 62.5 | 57.5 | 64.3 | 75.6 | 68.6 | 52.1 | 49.7 | 88.8 | 64.0 |
| BLIP-2 | 83.6 | **79.6** | 70.2 | 68.7 | 68.0 | 65.6 | 84.4 | 63.2 | 62.6 | 58.7 | 96.0 | 74.0 |
| ALBEF$_{3M}$ | 77.0 | **77.4** | 64.2 | 66.9 | 53.3 | **77.0** | 71.0 | **55.6** | 57.9 | 57.7 | 95.3 | 69.4 |
| REALBEF$_{3M}$ | **84.0** | 73.0 | 65.6 | 66.6 | 66.1 | 70.7 | 71.8 | 52.2 | 56.5 | 58.7 | 94.3 | 69.6 |
| X-VLM$_{3M}$ | 79.6 | 77.3 | 65.1 | 67.6 | 55.7 | 76.3 | **73.6** | 50.8 | **58.2** | 51.9 | **95.4** | 69.5 |
| REX-VLM$_{3M}$ | 82.6 | 76.9 | **66.6** | **69.9** | 67.1 | 76.4 | 69.9 | 52.6 | 55.5 | 65.4 | 95.2 | **70.9** |
| ALBEF$_{13M}$ | 75.4 | 78.0 | 68.1 | **70.7** | 68.0 | 76.4 | **74.8** | 55.7 | **60.6** | 61.5 | **96.1** | 72.2 |
| REALBEF$_{13M}$ | 74.9 | 77.8 | 67.7 | 68.8 | 64.1 | 75.9 | 72.7 | 53.0 | 56.4 | 55.8 | 95.2 | 70.4 |
| X-VLM$_{13M}$ | 74.7 | **79.2** | 65.4 | 69.2 | **73.2** | 75.7 | **74.8** | 53.5 | 54.8 | 51.9 | 95.8 | 71.3 |
| REX-VLM$_{13M}$ | **87.3** | 78.0 | **69.7** | 69.9 | 72.5 | **79.4** | 74.7 | **56.7** | 56.6 | 55.8 | 95.0 | **73.3** |

Table 5: Performance on the VALSE benchmark according to pairwise ranking accuracy. Best results are in **bold**. †**sns.** Counting small numbers. **adv.** Counting adversarial. **repl.** Action replacement. ‡ **Sp.rel.** Spatial relations.

| Model | Adjacency | Directional | Orientation | Projective | Proximity | Topological | Unallocated | Overall |
|---|---|---|---|---|---|---|---|---|
| Random | 50.0 / 50.0 | 50.0 / 50.0 | 50.0 / 50.0 | 50.0 / 50.0 | 50.0 / 50.0 | 50.0 / 50.0 | 50.0 / 50.0 | 50.0 / 50.0 |
| BLIP-2 | 59.8 / 54.9 | 50.0 / 43.3 | 52.5 / 57.1 | 59.8 / 63.6 | 56.2 / 51.2 | 66.4 / 67.0 | 75.0 / 66.7 | 61.2 / 61.5 |
| ALBEF$_{3M}$ | 54.5 / 55.6 | 45.5 / 42.2 | **67.8** / 56.2 | 64.2 / 62.7 | 56.2 / 52.0 | 69.8 / 65.0 | 71.9 / 47.1 | 63.7 / 60.1 |
| REALBEF$_{3M}$ | 55.3 / 52.8 | 54.5 / **48.9** | 64.4 / 53.6 | 66.6 / 64.0 | 59.4 / 54.5 | 66.4 / 62.9 | 68.8 / 60.8 | 64.0 / 60.2 |
| X-VLM$_{3M}$ | 56.1 / 54.9 | 50.0 / 43.3 | 64.4 / **57.1** | 63.0 / **66.6** | 60.9 / **55.3** | **69.5** / **66.0** | 68.8 / 56.9 | 63.5 / **62.3** |
| REX-VLM$_{3M}$ | **59.8** / **58.1** | **56.8** / **48.9** | 59.3 / 55.4 | **67.1** / 65.5 | 56.2 / **55.3** | 67.8 / 62.4 | **75.0** / **72.5** | **65.0** / 61.8 |
| ALBEF$_{13M}$ | 54.5 / 56.7 | 45.5 / 42.2 | 61.0 / **57.1** | 61.1 / 60.5 | 57.8 / 51.2 | 64.1 / 64.6 | 65.6 / 51.0 | 60.4 / 59.4 |
| REALBEF$_{13M}$ | 56.8 / 56.7 | **54.5** / 43.3 | 66.1 / **57.1** | 64.8 / 65.2 | **64.1** / 52.8 | 66.8 / 64.0 | **87.5** / 56.9 | 64.6 / 61.3 |
| X-VLM$_{13M}$ | 56.8 / 58.8 | 45.5 / 47.8 | 66.1 / **57.1** | 61.9 / 61.2 | 56.2 / 55.3 | 64.1 / 64.5 | 62.5 / **56.9** | 61.1 / 60.5 |
| REX-VLM$_{13M}$ | **67.4** / **60.2** | 50.0 / 47.8 | **67.8** / 53.6 | **68.4** / **67.0** | 60.9 / **56.1** | 72.5 / 67.3 | 75.0 / 51.0 | **68.4** / **63.5** |

Table 6: Dev/Test results on the VSR Random dataset. Best results are in **bold**.

| Model | Subj. | Verb | Obj. | Avg. |
|---|---|---|---|---|
| Random | 50.0 | 50.0 | 50.0 | 50.0 |
| CLIP (ViT-B/32) | 83.6 | 79.0 | 88.1 | 81.6 |
| BLIP-2 | 87.6 | 84.6 | 91.7 | 86.5 |
| ALBEF$_{3M}$ | 87.3 | 84.6 | 92.2 | 86.6 |
| REALBEF$_{3M}$ | 87.8 | 83.5 | **93.0** | 86.2 |
| X-VLM$_{3M}$ | **88.8** | **85.3** | 92.3 | **87.3** |
| REX-VLM$_{3M}$ | 88.2 | 85.2 | 92.8 | **87.3** |
| ALBEF$_{13M}$ | 86.9 | 84.9 | 92.0 | 86.7 |
| REALBEF$_{13M}$ | 88.8 | 85.2 | **93.4** | 87.5 |
| X-VLM$_{13M}$ | 88.1 | 85.5 | 92.3 | 87.3 |
| REX-VLM$_{13M}$ | **89.1** | **86.1** | 93.3 | **88.1** |

Table 7: Performance on the SVO-Probes benchmark according to pairwise ranking accuracy. Best results are in **bold**.

(+1.9/1.4pp for subject/object understanding). However, these improvements are smaller when training on 3M images, likely due to our relation-enhanced models requiring longer training to achieve top performance (see App. B.2). Overall, we note that *verb* understanding is still the most challenging aspect of SVO-Probes and that relation-enhanced models improve less for this subtask.

## B.2 Pretraining Dynamics

Bugliarello et al. (2023) showed that current, strong models achieve peak performance on different fine-grained tasks at different stages of pretraining. This motivates us to study the pretraining dynamics of our models. Performance for models pretrained on 3M and 13M images is shown in Figures 8 and 9.

We see that models performance, especially of our coarse-grained baselines, tends to fluctuate considerably on VSR tasks. For instance, X-VLM$_{3M}$'s accuracy on VSR Dev decreases during pretraining. Looking at relation-enhanced models, we find that they benefit from more training steps than the baselines. For instance, when pretrained on 3M images, they achieve peak fine-grained results after 350K–400K steps, while ALBEF$_{3M}$ and X-VLM$_{3M}$ do so within 200K steps (which is where we evaluate our models in Section 4). This is even more relevant when pretraining on 13M images, where our baselines' performance starts dropping after 250K steps, while our models are still improving at 500K steps. Longer pretraining and designing better schedules that balance coarse- and fine-grained tasks, and the different subtasks are promising di-

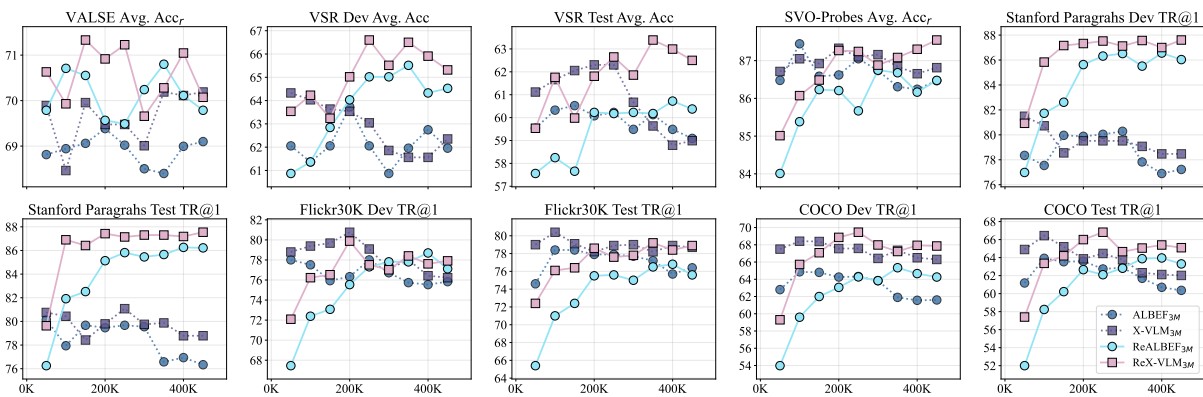

Figure 8: Pretraining dynamics of our models when learning from 3M images.

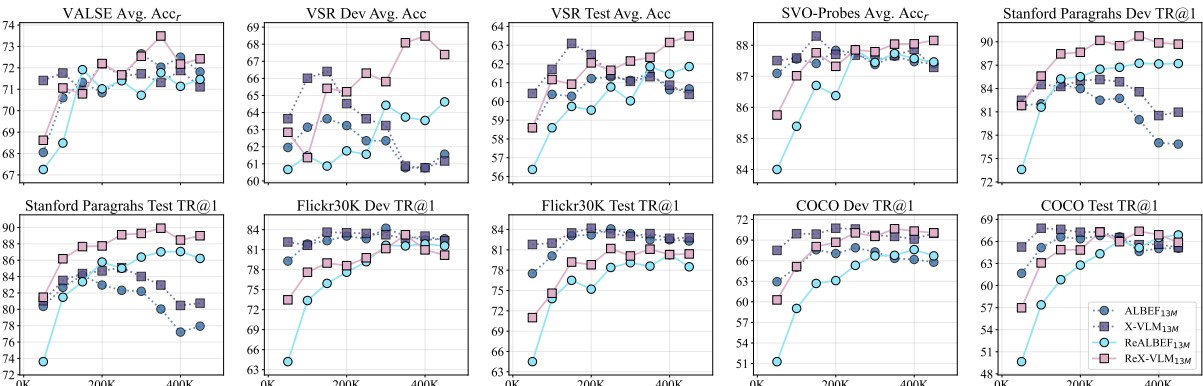

Figure 9: Pretraining dynamics of our models when learning from 13M images.

rections for future work to obtain a single checkpoint that performs well on both types of tasks.

Finally, Figures 12 to 15 show performance for our two proposed approaches when applied independently and together during pretraining of AL-BEF and X-VLM models on 3M and 13M images. On VSR, relation-enhanced models generally reach peak performance when combining both VSG and MRC. On VALSE, their performance degrades with respect to the baselines when using VSG alone. Moreover, looking at coarse-grained retrieval tasks throughout pretraining, we see that VSG degrades performance whist MRC can achieve on par or superior performance than the baselines. Interestingly, when combined, the final performance is closer to the stronger MRC objective.

## B.3 Checkpoint Selection Strategies

As discussed in Section 5 and shown in Figures 8 and 9, there is difference in convergence rates between relation-enhanced models and coarse-grained ones, with our models often requiring more steps to achieve peak performance. Here, we aim at complementing our discussion from Section 5.

Table 8 lists the performance of our models when performing checkpoint selection based on COCO Dev TR@1. Figure 10 lists the individual gains/losses of our models on each evaluation task according to different checkpoint selection strategies, when comparing against the standard approach of using the last checkpoint (200K steps for models trained on 3M images, and 500K steps for models trained on 14M images). Finally, Figure 11 reports the Spearman rank correlation coefficients between COCO Dev TR@1 and the other strategies, across all models. Here, the typical high coefficients indicate that the order with which models are ranked for a given task according to any strategy is mostly the same. That is, our findings from Section 4 hold regardless of the chosen checkpoint selection strategy.

| Model | VSR Random Dev / Test Acc | VALSE $Acc_r$ | SVO-Probes $Acc_r$ | Stanford Paragraphs IR@1/5 | TR@1/5 | Flickr30K IR@1/5 | TR@1/5 | COCO IR@1/5 | TR@1/5 |
|---|---|---|---|---|---|---|---|---|---|
| CLIP$_{400M}$ | N/A† | 64.0 | 81.6 | 45.3 / 73.1 | 53.4 / 80.1 | 88.0 / 98.7 | 68.7 / 90.6 | 58.4 / 81.5 | 37.8 / 62.4 |
| BLIP-2$_{129M}$ | 61.2 / 61.5 | 74.0 | 86.5 | 83.4 / 95.2 | 81.1 / 94.3 | 95.5 / 99.9 | 86.7 / 97.1 | 80.7 / 94.7 | 64.2 / 85.2 |
| ALBEF$_{3M}$ | 61.4 / 60.3 | 68.9 | 87.4 | 77.9 / 94.5 | 79.8 / 94.7 | 78.4 / 93.5 | 61.2 / 84.6 | 63.9 / 86.5 | 47.4 / 74.9 |
| REALBEF$_{3M}$ | 65.5 / 60.2 | 70.8 | 86.7 | 85.7 / 97.6 | 85.8 / 97.3 | 76.5 / 93.0 | 61.7 / 84.0 | 63.9 / 86.7 | 47.3 / 74.8 |
| X-VLM$_{3M}$ | 64.0 / 61.7 | 68.5 | 87.0 | 80.4 / 94.9 | 80.2 / 95.0 | 80.4 / 94.8 | 63.7 / 86.4 | 66.4 / 87.8 | 49.1 / 76.3 |
| REX-VLM$_{3M}$ | 66.6 / 62.6 | 71.2 | 87.2 | 87.1 / 98.0 | 88.3 / 97.4 | 77.6 / 94.4 | 61.8 / 84.3 | 66.8 / 88.2 | 49.4 / 76.2 |
| ALBEF$_{13M}$ | 62.4 / 61.3 | 71.4 | 87.7 | 82.3 / 96.1 | 82.6 / 95.8 | 84.1 / 94.6 | 67.3 / 87.6 | 66.8 / 87.9 | 49.7 / 76.5 |
| REALBEF$_{13M}$ | 63.5 / 61.5 | 71.1 | 87.7 | 87.1 / 97.5 | 86.3 / 97.2 | 80.3 / 93.6 | 64.9 / 85.6 | 66.5 / 87.9 | 48.8 / 75.9 |
| X-VLM$_{13M}$ | 64.5 / 62.5 | 72.2 | 87.7 | 84.7 / 96.8 | 85.4 / 96.3 | 84.2 / 96.6 | 68.0 / 87.8 | 67.3 / 88.4 | 50.7 / 76.9 |
| REX-VLM$_{13M}$ | 68.1 / 62.5 | 73.5 | 88.0 | 89.9 / 97.8 | 88.7 / 98.0 | 81.1 / 94.8 | 65.7 / 87.3 | 67.4 / 89.0 | 50.3 / 77.3 |

Table 8: Overall results on fine- (left) and coarse-grained (right) benchmarks. Models are evaluated at best COCO Dev TR@1. Values underlined in green (red) denote gains (losses) of relation-enhanced models on their baselines. †CLIP cannot be directly evaluated on VSR since it requires true/false predictions for a given image–text input, while CLIP is only trained with a contrastive loss. Best results are in **bold**.

Figure 10: Performance differences ($y$-axis) with respect to fixed checkpoints for all models according to different checkpoint selection tasks ($x$-axis).

Figure 11: Spearman rank correlation coefficients of different checkpoint selection tasks ($x$-axis) with using COCO Dev TR@1 for our evaluation tasks ($y$-axis).

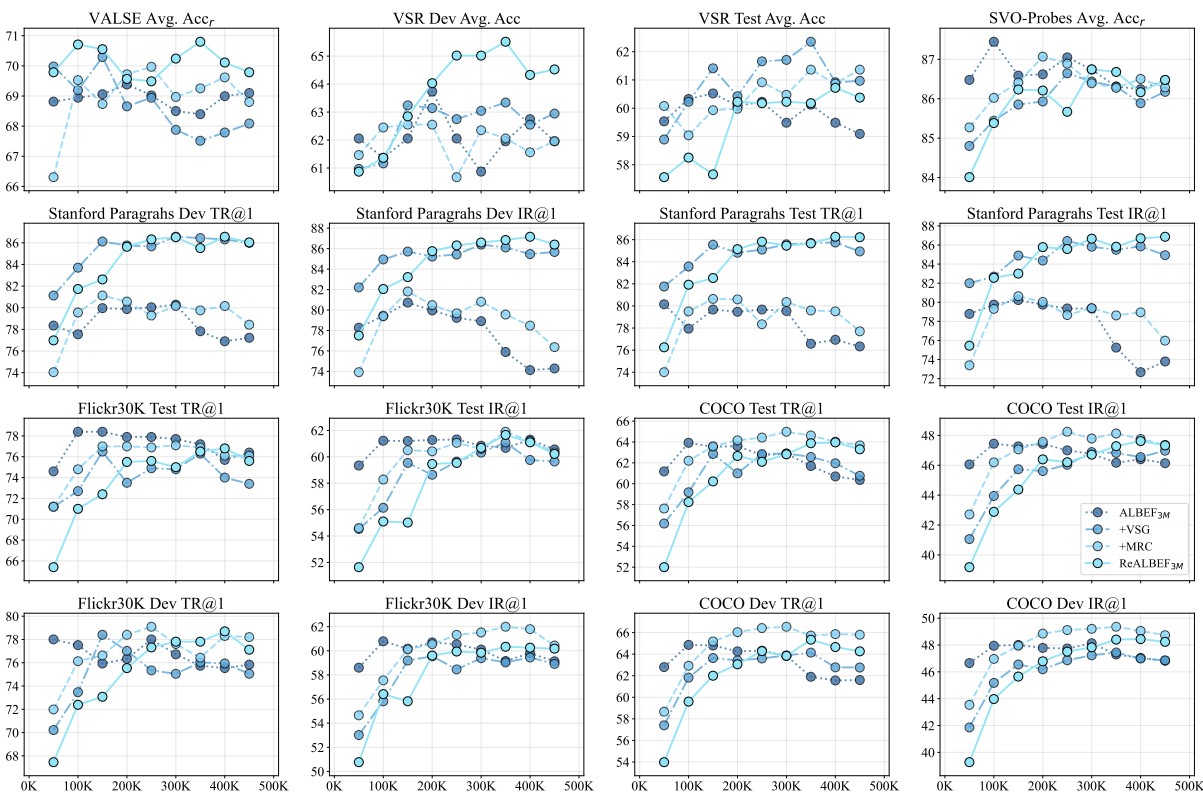

Figure 12: Pretraining dynamics of our approaches on ALBEF models pretrained on 3M images.

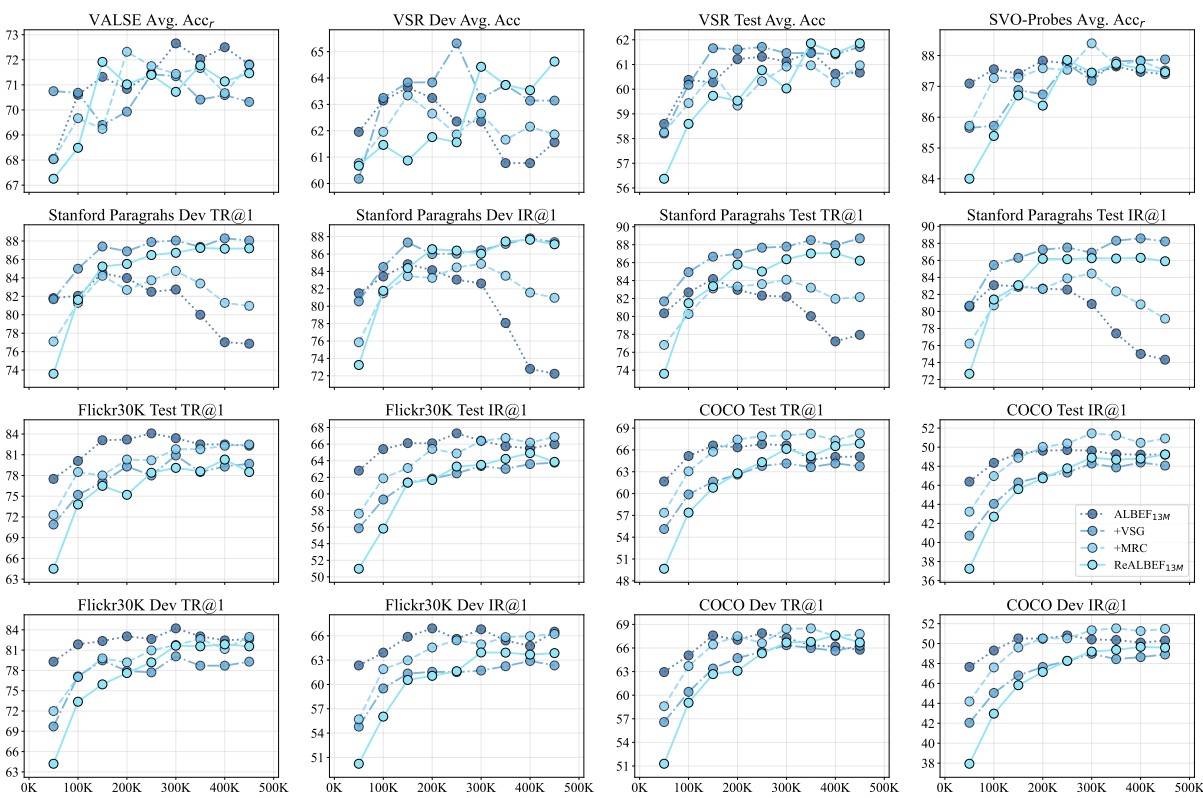

Figure 13: Pretraining dynamics of our approaches on ALBEF models pretrained on 13M images.

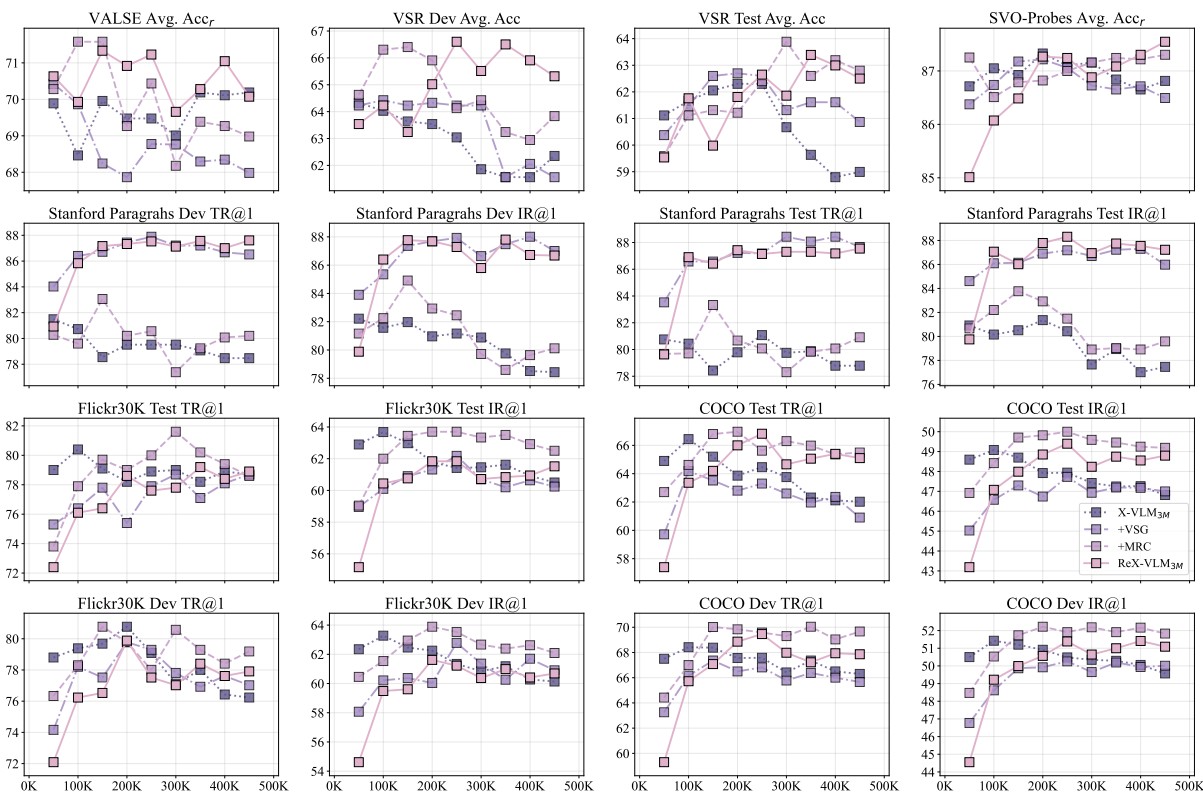

Figure 14: Pretraining dynamics of our approaches on X-VLM models pretrained on 3M images.

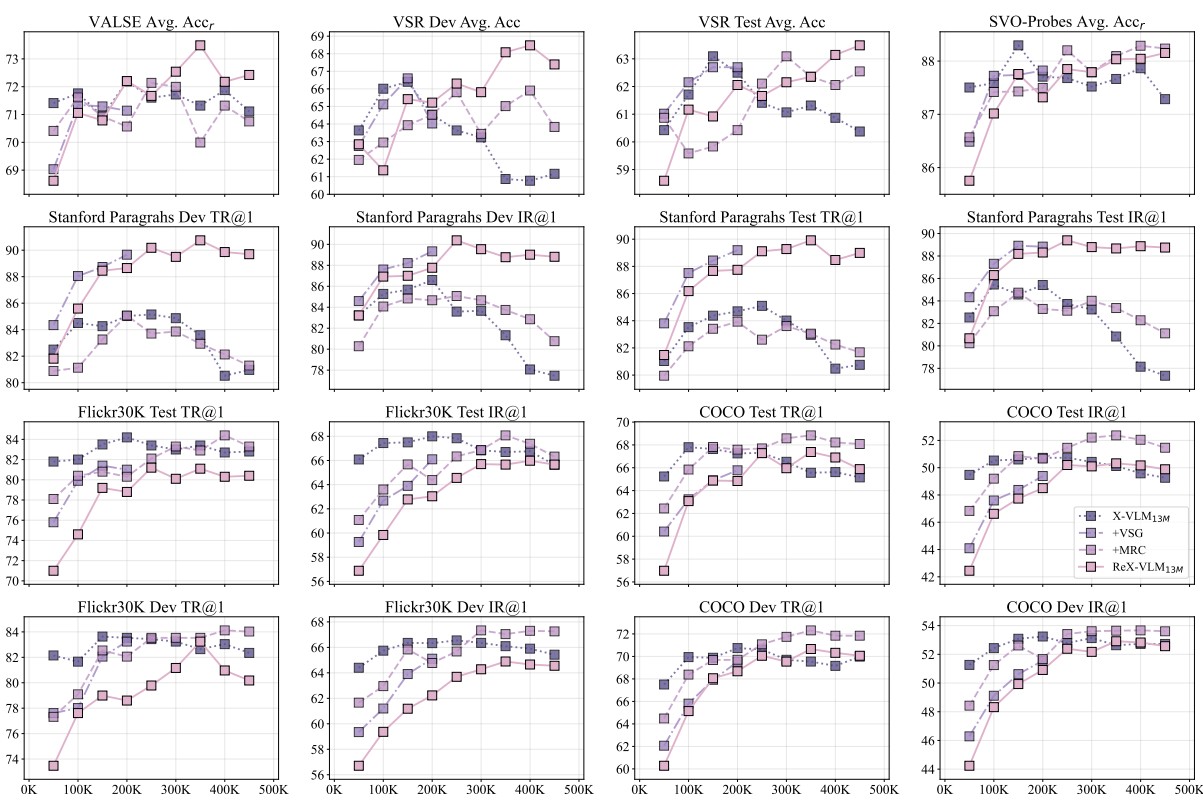

Figure 15: Pretraining dynamics of our approaches on X-VLM models pretrained on 13M images.