# OpenReview forum: "Weakly-Supervised Learning of Visual Relations in Multimodal Pretraining"
_EMNLP/2023/Conference — EMNLP 2023 Main_

### Official Review · Reviewer_y9T8 · 2023-08-10

**Typos Grammar Style And Presentation Improvements:** Line 614 in the section of Limitation…
**Soundness:** 4

**Excitement:**

3: Ambivalent: It has merits (e.g., it reports state-of-the-art results, the idea is nice), but there are key weaknesses (e.g., it describes incremental work), and it can significantly benefit from another round of revision. However, I won't object to accepting it if my co-reviewers champion it.

**Paper Topic And Main Contributions:**

This paper introduces supervised visual relation data into the process of multimodal pre-training to enhance the fine-grained multimodal understanding ability of VLMs. Specifically, the pre-training is divided into two parts: 1) *verbalized scene graphs* applies the standard multimodal pre-training objectives used on image captions to verbalized scene graph annotations; 2) *masked relation classification* predicts the relation from a pre-defined relation vocabulary between two entities. Experimental results show the proposed method can improve the VLMs' visual spatial reasoning and fine-grained understanding capabilities.

**Questions For The Authors:**

- The relative decrease in the performance of REALBEF 3M compared to ALBEF 3M across most coarse-grained tasks suggests that the performance of the method in this paper might be influenced by the model architecture and scale. How do you view this phenomenon?

**Reasons To Accept:**

- By adding masked relation classification loss in multimodal pre-training, the proposed method achieves state-of-the-art performance on diverse fine-grained benchmarks.
- The proposed method achieves significant improvements with a relatively small amount of data.
- The paper is well-written. It seems that experiments and analysis are comprehensive and the figures and tables are visually appealing.

**Reasons To Reject:**

- The experiments primarily focus on tasks related to scene graphs, which results in limited coverage of other multi-modal tasks.
- The baseline and backbone models might be somewhat outdated, given the current proliferation of multi-modal models based on LLaMA.
- The connection between weak supervision and the motivation behind this paper is not very close as the proposed method distinctly differs from some traditional understandings of weakly supervised approaches. A more accurate description might be a multi-modal pretraining method suited for *low-resource scenarios*.

**Reproducibility:**

4: Could mostly reproduce the results, but there may be some variation because of sample variance or minor variations in their interpretation of the protocol or method.

**Reviewer Confidence:**

4: Quite sure. I tried to check the important points carefully. It's unlikely, though conceivable, that I missed something that should affect my ratings.

---

> ### Author Rebuttal · Authors · 2023-08-28
>
> Thank you for carefully reviewing our paper! We’re glad that you found it well-written, with comprehensive experiments and analysis that support state-of-the-art performance on diverse fine-grained benchmarks from a relatively small amount of data.
>
> > The experiments primarily focus on tasks related to scene graphs, which results in limited coverage of other multi-modal tasks.
>
> We note that none of the tasks that we evaluate models on require scene graph understanding or generation. We hypothesised (and later verified) that successful modelling of scene graphs could lead to improved visual relation understanding on VSR. Moreover, we show that our methods improve general fine-grained vision-and-language understanding, while keeping competitive performance on coarse-grained tasks. The fine-grained tasks that we benchmark on cover a wide range of multimodal abilities. For instance, SVO-Probes measures verb understanding, and VALSE covers six different linguistic phenomena: existence, counting, plurality, relations, actions and coreference. We provide a breakdown of performance across each subtask in App B.1.
>
> > The baseline and backbone models might be somewhat outdated, given the current proliferation of multi-modal models based on LLaMA.
>
> We would like to point out that Bugliarello+(ACL’23) showed that X-VLM is the current state-of-the-art model for fine-grained understanding, outperforming LLM-based models like Flamingo and BLIP-2. As such, we simply reported the performance of the best *concurrent* work (BLIP-2; ICML’23) as a reference for the reader.
>
> We also want to stress that multimodal extensions of LLaMa [1, 2] are *concurrent* work and, according to *ACL policies (last bullet point of https://www.aclweb.org/adminwiki/index.php/ACL_Policies_for_Submission,_Review_and_Citation) lack of comparison against them should not be considered a valid reason for rejection.
>
> Moreover, our work aims at improving strong multimodal models. As Reviewer VUQu stressed, our VSG data-to-text strategy is versatile, and it can seamlessly be applied to LLaMA-based models too, as well as any other large model.
>
> Having said that, if you believe that adding a specific baseline would be a nice-to-have addition to our results, we will include it when we revise our paper should it get accepted.
>
> > The connection between weak supervision and the motivation behind this paper is not very close as the proposed method distinctly differs from some traditional understandings of weakly supervised approaches. A more accurate description might be a multi-modal pretraining method suited for low-resource scenarios.
>
> We note that, unfortunately, the term “weakly-supervised learning” is overloaded. Our usage aligns with one type of weak supervision: incomplete supervision, defined in [3] as “learning where only a subset of training data is given with labels.” We refer to [3] for a relevant review of weakly-supervised learning.
>
> If you and other reviewers, however, think that the title is misleading, we’ll consider updating it accordingly.
>
> > The relative decrease in the performance of REALBEF 3M compared to ALBEF 3M across most coarse-grained tasks suggests that the performance of the method in this paper might be influenced by the model architecture and scale. How do you view this phenomenon?
>
> Good observation! We find that this is due to suboptimal data mixing. In fact, as shown in Figure 9 and 12 in the Appendix, ReALBEF_3M matches or outperforms ALBEF_3M when trained for more steps. We will revise our Conclusion to highlight this point, and encourage future work to investigate this multi-task learning problem further.
>
> > Line 614 in the section of Limitations has a typo "??".
>
> Thank you, we’ll fix it!
>
> > Excitement
>
> We hope that our response helped clarify your concerns. (1) None of our evaluation tasks is related to scene graphs. (2) LLaMa-based models are concurrent to us, and do not present any distinctive fine-grained abilities. (3) We use “weakly-supervised learning” in a similar way as the literature. If you no longer think that our paper would “significantly benefit from another round of revision,” we’d be thrilled if you considered increasing your score accordingly.
>
> We’ll be happy to engage further during the discussion phase, thank you!
>
> ---
>
> [1] Zhang et al. LLaMA-Adapter: Efficient Fine-tuning of Language Models with Zero-init Attention. (2023). arXiv preprint 2303.16199
>
> [2] Gao et al. LLaMA-Adapter V2: Parameter-Efficient Visual Instruction Model. (2023). arXiv preprint 2304.15010
>
> [3] Zhou. A brief introduction to weakly supervised learning. (2018). National Science Review. https://doi.org/10.1093/nsr/nwx106

---

### Official Review · Reviewer_tnr9 · 2023-08-11

**Soundness:** 4

**Excitement:**

4: Strong: This paper deepens the understanding of some phenomenon or lowers the barriers to an existing research direction.

**Paper Topic And Main Contributions:**

This paper focuses on multimodal pretraining for fine-grained multimodal understanding.

**Reasons To Accept:**

This work is well-motivated and has extensive experiments to show the priority and effectiveness over several recent models. The proposed method is novel and effective. And the analysis of the experiments are detailed.

**Reasons To Reject:**

Visual representation is missing to demonstrate whether the model indeed possesses the ability for fine-grained comprehension.

**Reproducibility:**

4: Could mostly reproduce the results, but there may be some variation because of sample variance or minor variations in their interpretation of the protocol or method.

**Reviewer Confidence:**

3: Pretty sure, but there's a chance I missed something. Although I have a good feel for this area in general, I did not carefully check the paper's details, e.g., the math, experimental design, or novelty.

---

> ### Author Rebuttal · Authors · 2023-08-28
>
> Thank you for carefully reviewing our paper! We’re happy that you found our work well-motivated, with a detailed analysis, and with extensive experiments that show the effectiveness of our novel method.
>
> > Visual representation is missing to demonstrate whether the model indeed possesses the ability for fine-grained comprehension.
>
> This is a good point, thank you for raising it! We did not include any visual representations for two main reasons.
>
> First, VALSE and SVO-Probes are evaluated through foils, which contain only a single difference between the true and false pairs (e.g., only a word differs between true and false captions). That is, the only way that a model can succeed is indeed to correctly understand that difference. Hence, we felt that the quantitative results obtained through these fine-grained benchmarks were enough to highlight the fine-grained abilities gained by the models.
>
> Second, it is not trivial to demonstrate fine-grained comprehension in VSR and Stanford Paragraphs. For instance, VSR focuses on relation understanding (e.g., above, next to), and it is not clear to us what would be the expected behaviour (Should the model pay attention to both objects? Should it focus on the contact point?).
>
> Having said that, if you have specific suggestions, we’d be happy to include them in the extra page that we would be granted should the paper be accepted. Thank you!

---

### Official Review · Reviewer_VUQu · 2023-08-11

**Soundness:** 4

**Excitement:**

3: Ambivalent: It has merits (e.g., it reports state-of-the-art results, the idea is nice), but there are key weaknesses (e.g., it describes incremental work), and it can significantly benefit from another round of revision. However, I won't object to accepting it if my co-reviewers champion it.

**Paper Topic And Main Contributions:**

This paper introduces a weakly-supervised learning approach for enhancing visual relations in multimodal pretraining by leveraging verbalized scene graphs. To optimize the learning of visual relations, the method builds upon the loss function introduced by ALBEF, adapting it to harness the structured information from the verbalized scene graphs. Additionally, the research introduces a masked relation classification as a supplementary loss function to further improve the learning process.

The effectiveness of the proposed approach is empirically validated across various benchmark datasets. For coarse-grained retrieval, the experiments were conducted on Flickr30K and COCO datasets, while for fine-grained retrieval, datasets such as VSR, VALSE, SVO-Probes, and Stanford Paragraphs were used. The results section provides a comprehensive analysis, shedding light on the nuances of the task and highlighting the pivotal insights drawn from the experiments.

**Questions For The Authors:**

1. In section 2, what does l_{i} mean?

**Reasons To Accept:**

- Emphasizing visual relations is pivotal for a nuanced understanding in Visual Language Pretraining (VLP).
- Leveraging verbalized scene graphs proves effective during pre-training. This strategy is versatile, allowing its integration into other VLP models, as showcased in the adaptation of the ALBEF loss function in this research.
- The proposed methodology's efficacy is demonstrated through rigorous testing on several benchmark datasets. Coarse-grained retrieval assessments utilized the Flickr30K and COCO datasets. In contrast, fine-grained evaluations employed datasets like VSR, VALSE, SVO-Probes, and Stanford Paragraphs. The subsequent analysis in the results section offers valuable insights into the task's intricacies.

**Reasons To Reject:**

- Visual scene graphs have been widely used not only in the vision but visual-language domains.
- Without datasets, code of methods and checkpoints, it is hard to verify the results reported in the paper.

**Reproducibility:**

4: Could mostly reproduce the results, but there may be some variation because of sample variance or minor variations in their interpretation of the protocol or method.

**Reviewer Confidence:**

3: Pretty sure, but there's a chance I missed something. Although I have a good feel for this area in general, I did not carefully check the paper's details, e.g., the math, experimental design, or novelty.

---

> ### Author Rebuttal · Authors · 2023-08-28
>
> Thank you for carefully reviewing our paper! We’re happy that you found our analysis comprehensive, and our strategy to be versatile and effective, as demonstrated through rigorous testing on several benchmarks. We agree that emphasising visual relations is pivotal for a nuanced understanding, and we hope that others, like you, will gain valuable insights into the task's intricacies from our paper!
>
> > Visual scene graphs have been widely used not only in the vision but visual-language domains.
>
> We agree! We provided a few relevant citations for usage of scene graphs in L057-065, with further emphasis on vision-and-language domains in L522-544. However, they have been understudied for multimodal pretraining. This is where our work fills the gap. We demonstrate, for the first time, that adding scene graphs for a small number of images on top of millions of image–text pairs from the Web results in improved coarse-grained and fine-grained skills.
>
> We are only aware of two previous studies that investigated the use of scene graphs for vision-and-language pretraining. In addition to the core question that we investigate in this paper (intrinsic performance in both coarse- and fine-grained tasks), we highlight the difference in methodology in L530-544.
>
> If you have specific references that you think are missing, we’d be happy to integrate them in our revision.
>
> > Without datasets, code of methods and checkpoints, it is hard to verify the results reported in the paper.
>
> We use the same evaluation data as Bugliarello+(ACL’23), which is publicly available at https://github.com/e-bug/fine-grained-evals. The pretraining data is also publicly available, and it consists of the standard datasets that have been used by the community (COCO, SBU, Visual Genome, Conceptual Captions). The codes for ALBEF and X-VLM are also open source, and we think that the details in the paper (see App A) should be enough to reproduce our method. We would, of course, be happy to integrate any details that you think are missing. We will reply to any further inquiry by the community should the paper be accepted, and we will try to release our checkpoints.
>
> > In section 2, what does l_{i} mean?
>
> l_{i} is the label (e.g., car, duck) for the entity e_{i}. Thank you for pointing this out, we’ll make it clear when we revise our paper.
>
> > Reproducibility
>
> In addition to the clarification of datasets and code above, in App A, we meticulously made sure to provide information for all criteria listed in the EMNLP 2023 reproducibility checklist (https://2023.emnlp.org/calls/main_conference_papers/#reproducibility-criteria). We hope this weakens your concerns on being able to reproduce our results.
>
> > Excitement
>
> We hope that our response clarifies your concerns with the position and reproducibility of our study, and we’d be happy to engage further during the discussion phase. We are delighted that you found our work important and thorough. We did not find any major concerns and suggestions that would make our paper “significantly benefit from another round of revision.” So we hope you’ll consider increasing your score according to your overall recommendation, thank you!

---

### Official Review · Reviewer_JtYm · 2023-08-13

**Soundness:** 4

**Excitement:**

4: Strong: This paper deepens the understanding of some phenomenon or lowers the barriers to an existing research direction.

**Paper Topic And Main Contributions:**

This paper studies the effect of visual relation in multi-modal pretraining. More specifically, the authors propose a pre-training strategy to include the visual relation annotations together with natural language. The extensive experiments show that the proposed strategy can boost the VLM trained on vision-language annotation.

**Questions For The Authors:**

See the reasons to reject

**Reasons To Accept:**

1. The experiments are solid and extensive (although misses some discussions, which I will explained in the reasons to reject)

2. The contribution of this paper to scene graph area is important.

**Reasons To Reject:**

I want to see more discussions of ablation study. More specifically, I am very curious about the strategy design. Why do we pre-train the model in a weakly-supervised manner yet providing some prior information on the locations (the verbalized sentences are sorted based on locations). Would the authors like to explain why we remove the exact mapping information but reserve the location prior? I did not see the ablation study.

**Reproducibility:**

3: Could reproduce the results with some difficulty. The settings of parameters are underspecified or subjectively determined; the training/evaluation data are not widely available.

**Reviewer Confidence:**

2: Willing to defend my evaluation, but it is fairly likely that I missed some details, didn't understand some central points, or can't be sure about the novelty of the work.

---

> ### Author Rebuttal · Authors · 2023-08-28
>
> Thank you for carefully reviewing our paper! We’re glad to see that you found our experiments solid and extensive, and that our approach is important for the scene graph area. Indeed, we are the first to show the effectiveness of using scene graphs to improve fine-grained understanding in pretrained vision-and-language models (VLMs), and hope future work will investigate this further.
>
> > Why do we pre-train the model in a weakly-supervised manner yet providing some prior information on the locations (the verbalized sentences are sorted based on locations). Would the authors like to explain why we remove the exact mapping information but reserve the location prior?
>
> We are not sure we completely understand the ablation you are proposing, are you suggesting to use objects’ bbox coordinates and their relations in the same loss?
>
> If that’s the case, our MRC loss implicitly does that by visually masking two entities separately, and then trying to predict their relation (see Fig2 right), where the visual mask is computed from the objects’ bboxes.
>
> In the VSG loss, we simply use the bbox coordinates to order the visual content in a scene. Yet, our ablation in Fig5 shows that sorting based on bbox coordinates is not necessary to obtain strong representations. It is also true that we do not use such fine-grained mapping between the objects and their locations in the VSG loss (which is an interesting idea for follow-up work!). Still, we also remind that our main baseline model (X-VLM) already modelled bbox locations during pretraining.
>
> We investigate the benefits of learning visual relations between entities for fine-grained skills in VLMs. The comparisons between ALBEF and X-VLM models disentangles the information of modelling object locations. Interestingly, in L399-422, we find that modelling relations alone is more effective than object localisation when training VLMs at scale.
>
> Please let us know in the discussion phase if we misunderstood your concern, and we’d be happy to engage further to clarify and answer your questions.
>
> > Reproducibility
>
> In App A, we meticulously made sure to provide information for all criteria listed in the EMNLP 2023 reproducibility checklist (https://2023.emnlp.org/calls/main_conference_papers/#reproducibility-criteria). The codes for ALBEF and X-VLM are also open source. We use the same evaluation data as Bugliarello+(ACL’23), which is publicly available at https://github.com/e-bug/fine-grained-evals. The pretraining data is also publicly available and it consists of standard datasets that have been used by the community (COCO, SBU, Visual Genome, Conceptual Captions). We hope this resolves your concerns and are happy to provide additional details if needed.
>
> > Excitement
>
> We hope that our response addresses your concerns with our study, though we would also appreciate clarification on how our work will "significantly benefit from another round of revision" that is reflected in the excitement score. From our understanding, the main concern is a lack of an ablation (which we think is not a “key weakness” that motivates a score of 3). We are happy to include additional missing ablations in the extra page if you can clarify what is needed.

---

### Meta-Review · Area_Chair_Yn6x · 2023-09-23

**Recommendation:** 5

**Metareview:**

The paper focuses on enhancing fine-grained multimodal understanding through a weakly-supervised learning approach that incorporates visual relation annotations into multimodal pretraining. The method introduces a novel strategy, including verbalized scene graphs and masked relation classification, and demonstrates its effectiveness through extensive experiments on benchmark datasets. Despite some minor concerns, reviewers unanimously praised the soundness of this work with excitement. Overall, the paper makes a valuable contribution to the field of multimodal pretraining, and can benefit from addressing the limitations and providing a more extensive discussion of its applicability to various multimodal tasks in the camera ready version.

---

### Decision · Program_Chairs · 2023-10-07

**Decision:**

Accept-Main

**Comment:**

The paper focuses on enhancing fine-grained multimodal understanding through a weakly-supervised learning approach that incorporates visual relation annotations into multimodal pretraining. The method introduces a novel strategy, including verbalized scene graphs and masked relation classification, and demonstrates its effectiveness through extensive experiments on benchmark datasets. Despite some minor concerns, reviewers unanimously praised the soundness of this work with excitement. Overall, the paper makes a valuable contribution to the field of multimodal pretraining, and can benefit from addressing the limitations and providing a more extensive discussion of its applicability to various multimodal tasks in the camera ready version.